# Interventions to improve adherence/ compliance to home noninvasive positive pressure ventilation in stable hypercapnic chronic obstructive pulmonary disease patients: A systematic review protocol

**Amani Kacem[1,2◉], Imen Ben Saida [2,3◉]\*, Dhekra Chebil[2,4], Helmi Ben Saad[2,5‡], Mohamed Boussarsar [2,3‡]**

**1** Pulmonology Department, Ibn El Jazzar University Hospital, Kairouan, Tunisia, **2** Faculty of Medicine of Sousse, University of Sousse, Sousse, Tunisia, **3** Medical Intensive Care Unit, Research Laboratory LR12SP09 "Heart Failure", Farhat Hached University Hospital, Sousse, Tunisia, **4** Preventive Medicine Department, Ibn Al Jazzar University Hospital, Kairouan, Tunisia, **5** Laboratory of Physiology and Functional Explorations, Research Laboratory LR12SP09 "Heart Failure", Farhat Hached University Hospital, Sousse, Tunisia

◉ These authors contributed equally to this work.
‡ These authors are joint senior authors on this work.
\* imen.bensaida@yahoo.com

## Abstract

### Introduction

Chronic obstructive pulmonary disease (COPD) is a prevalent condition often leading to chronic hypercapnic respiratory failure in its advanced stages. Home noninvasive positive pressure ventilation (Home-NIPPV) has emerged as a key therapeutic strategy for managing stable hypercapnic COPD patients, improving survival rates, and enhancing quality of life. Despite these benefits, patient adherence/compliance to Home-NIPPV remains a significant challenge, hindered by various barriers. The present paper is a protocol of a systematic review that aims to identify and evaluate interventions designed to improve adherence/compliance to Home-NIPPV in stable hypercapnic COPD patients.

### Methods

The protocol is developed following the Preferred Reporting Items for Systematic Reviews and Meta-Analyses Protocols guidelines and was registered in PROSPERO (CRD42024581616). A comprehensive literature search across *PubMed*, *Scopus* and *Cochrane Library* will be conducted. Eligible studies will include randomized controlled trials that focus on interventions that aims to improve adherence/compliance to Home-NIPPV in stable hypercapnic COPD patients. Data extraction will be meticulously carried out by two independent reviewers. The Joanna Briggs Institute critical appraisal tool will be utilized to assess the quality and risk of bias of the included studies. The findings of this systematic review will be synthesized to provide a thorough understanding of effective

**Data availability statement:** No datasets were generated or analysed during the current study. All relevant data from this study will be made available upon study completion.

**Funding:** The author(s) received no specific funding for this work.

**Competing interests:** The authors have declared that no competing interests exist.

strategies to enhance adherence/compliance to Home-NIPPV in stable hypercapnic COPD patients.

## Conclusion

The results of this review could inform clinical practice and guide the development of targeted strategies to improve adherence/compliance to Home-NIPPV and consequently outcomes in stable hypercapnic COPD patients.

## Registration

This protocol was registered in the International Prospective Register of Systematic Reviews under the reference code CRD42024581616.

## Introduction

Chronic obstructive pulmonary disease (COPD) is a progressive lung disease characterized by persistent airflow obstruction, recurrent exacerbations, and declining lung function, placing a substantial burden on healthcare systems [1]. Globally, it is considered the third-most common cause of death [2]. Home noninvasive positive pressure ventilation (Home-NIPPV) has become a well-established treatment for chronic hypercapnic COPD patients, improving their morbidity, mortality, and quality of life [3,4]. However, adherence/compliance (*i.e.,*; voluntary cooperation of the patient in following a prescribed regimen, including timing, dosage, frequency, and consistency) to Home-NIPPV is crucial for achieving optimal therapeutic benefits and reduces the risk of complications or disease progression [5,6].

Despite its importance, adherence/compliance rates vary widely, and non-adherence remains a significant challenge in hypercapnic COPD patients' management [7,8]. The process behind acceptance and adherence/compliance to Home-NIPPV is dynamic and nonlinear, influenced by positive and negative experiences, facilitators, and barriers [9]. In fact, some patients may refuse Home-NIPPV, while others may initially accept it but later show transient daily compliance without achieving full, sustained adherence. A few, however, will maintain it over a prolonged period, eventually developing consistent adherence [9]. In the context of Home-NIPPV, barriers to adherence/compliance are multifactorial, including patient-related, device-related, and support service-related factors [9,10]. The optimal location for initiating therapy, the most effective Home-NIPPV modes and settings, and the best monitoring strategies to improve Home-NIPPV adherence/compliance in stable hypercapnic COPD patients remain uncertain, as reflected in current European Respiratory Society (ERS) and the American Thoracic Society (ATS) guidelines [3,11–13]. Understanding and addressing adherence/compliance issues is essential for optimizing the efficacy of this highly effective technique [14]. While numerous interventions have been studied [15–22], the relative efficacy of these interventions remains unclear.

To address the aforementioned knowledge gap, the present paper is a protocol of a systematic review that aims to identify and evaluate interventions designed to improve adherence/compliance to Home-NIPPV in stable hypercapnic COPD patients.

## Methods

### Review question

In stable hypercapnic COPD patients, which interventions are effective at improving adherence/compliance with Home-NIPPV?

## Study design

This protocol was registered in the International Prospective Register of Systematic Reviews under the reference code CRD42024581616. *Cochrane*, *Prospero* (Prospective Register of Systematic Reviews), Open Science Framework, and *PubMed* were first searched on August 20, 2024, to confirm that no similar systematic review was already registered. This study will be reported following the guidelines outlined in the "Preferred Reporting Items for Systematic reviews and Meta-analyses [23]. (S1Checklist)

## Eligibility criteria

The eligibility criteria for study inclusion will be established based on the PICOS framework [24], encompassing the following:

- **Population (P):** Adult patients aged ≥ 18 years with confirmed diagnosis of COPD with indication of Home-NIPPV for chronic hypercapnic respiratory failure, free from neuromuscular diseases; and under no mechanical ventilation through a tracheostomy.

- **Intervention (I):** Any intervention designed to improve stable hypercapnic COPD patients' adherence/compliance to Home-NIPPV, targeting patient-related, device-related, and support service-related factors (***(e.g.,*** place of initiation, ventilation mode, type, and settings, adjuvant therapies and different types of monitoring…).

- **Comparison (C):** Comparative groups will include intervention groups compared to either a control group receiving no intervention or a sham/placebo intervention, or interventions compared to standard care, or trials evaluating different intervention approaches.

- **Outcomes (O):** The primary outcomes will be Home-NIPPV adherence/compliance.

- **Study design (S):** Will be included randomized controlled trials (RCTs) or crossover randomized trials reporting interventions to improve adherence/compliance with Home-NIPPV in stable hypercapnic COPD patient. Only studies published in English language with adherence or compliance as primary or secondary outcomes will be included. No constraints will be applied regarding the study setting, country of origin, or publication period. Non-randomized study designs, observational studies, reviews, editorials, qualitative papers, letters to editors, commentaries, case series, and case reports will be excluded from consideration. Studies evaluating adherence/compliance to home Continuous Positive Airway Pressure or home High Flow Nasal Cannula will be excluded. Studies including patients with restrictive diseases will be excluded.

## Search Strategy

A comprehensive literature search will be conducted online, utilizing three reputable databases, *PubMed*, *Scopus,* and *Cochrane Library*. The search will be updated to make sure all relevant articles are included before finalizing the study results. For *PubMed*, the search strategy will involve the utilization of a combination of four specific Medical Subject Headings terms: ((("Patient Compliance") OR "Treatment Adherence and Compliance") AND "Pulmonary Disease, Chronic Obstructive") AND "Noninvasive Ventilation". In the case of *Scopus* and *Cochrane Library*, the same terms will be sought in the article titles, abstracts, and/or keywords. To enhance search comprehensiveness, respective references' lists of the included studies will be meticulously examined for additional relevant literature.

## Study selection

The search results from each of the two databases will be consolidated, and duplicates will be removed using the EndNote 20 library. In the screening phase, two independent investigators (*AK* and *IBS* in the authors' list) will independently review the titles and abstracts identified through the search strategy to assess their relevance to the topics under investigation. Any disagreements regarding study eligibility will be resolved through consensus between the two investigators, and if necessary, by consulting a third party (*HBS* in the authors' list). Following this, a thorough review of the full texts of studies that meet the predefined eligibility criteria will be conducted to confirm their inclusion in the systematic review.

## Data extraction

Data extraction will be performed collaboratively by two investigators (*AK* and *IBS* in the authors' list) via a standardized format using Microsoft Excel software. Data extracted from each article will include:

- Detailed information on methodological characteristics of the retained studies ((**e.g.,** first author, aim(s) related to the study proposes, study design, study period, country, sample size calculation, blinding technique, randomization method).

- Recruitment methods, inclusion, non-inclusion and exclusion criteria.

- Characteristics of COPD participants: Sample size, age, sex ratio, body mass index, global initiative for chronic obstructive lung disease stage, daytime arterials partial pressure of carbon dioxide ($PaCO_2$) in stable condition, comorbidities, smoking status (pack/year), inhaler regimen/medications, number of exacerbations within the last year, number of hospitalizations during the last year, marital status, educational level, naïve or familiar with Home-NIPPV.

- Home-NIPPV procedures description: Place of initiation, type of the mask ((**e.g.,** nasal, oral-nasal or full-face), ventilator mode and settings, adjuvant therapies and different types of monitoring, home oxygen therapy, prescribed Home-NIPPV usage time, acceptance of Home-NIPPV rate.

- Data from ventilator downloads: adherence/compliance rate, mean daily use hours, timing of evaluation, number of hours of use per night, percentage of time spent on Home-NIPPV compared to prescribed time, frequency of interruptions, exhaled tidal volume, percent triggered breaths, leaks, residual apnea-hypopnea index, patient-ventilator asynchronies.

- Provided definition of acceptance, compliance, and adherence to Home-NIPPV.

- Outcomes: Hypercapnic encephalopathy, sleep quality, improvement of $PaCO_2$, residual respiratory events, readmission rate, and mortality rate.

The extracted data will undergo a comprehensive verification process by a third investigator (*MB* in the authors' list) to confirm its reliability and completeness. The discrepancies during the data collection and extraction phases will be addressed through in-depth discussions.

## Risk of bias (quality) assessment

To assess the methodological quality of included studies, the Joanna Briggs Institute (JBI) critical appraisal tool will be used [25]. This tool evaluates various aspects of study design, with items rated as "yes," "no," "unclear," or "not applicable." To enhance reliability, two

investigators (*AK* and *IBS* in the authors' list) will independently apply the JBI checklist to each study. Discrepancies will be resolved through consensus, with a third investigator (*DC* in the authors' list) mediating if necessary. A percentage score will be calculated based on the number of affirmative responses to checklist items. This checklist includes 13 items ([Fig 1]). Studies will be categorized as having high, moderate, or low risk of bias according to predetermined thresholds: $\leq 49\%$, 50–69%, and $\geq 70\%$ affirmative responses, respectively [26].

## Data synthesis

A qualitative synthesis will provide a narrative overview of the included studies, with a focus on interventions aimed at enhancing adherence or compliance to home NIPPV in stable hypercapnic COPD patients. The methodological characteristics of RCTs included (e.g., study design, randomization method, sample size, blinding techniques), participant recruitment processes, inclusion/exclusion criteria, and detailed COPD patient profiles such as age, sex ratio, body mass index, disease severity, comorbidities, smoking status, and prior experience with Home-NIPPV will be summarized. The synthesis will further describe the details of interventions, including Home-NIPPV procedures, ventilator type, ventilation settings, adherence rates, mean daily usage hours, and monitoring methods. Reported definitions of acceptance, compliance, and adherence will be summarized to highlight variability across studies. Reported adherence/compliance related outcomes, such as improvements in $PaCO_2$, sleep quality, readmission rates, and mortality, will be synthesized to assess the overall impact of interventions.

The narrative synthesis will identify patterns, similarities, and differences across the included studies. The results will be discussed in relation to the objectives of this review considering the quality of the studies, the consistency of the findings, and potential biases, and will be clearly presented using tables and graphs.

| Items (I) |
|---|
| I 1. Was true randomization used for assignment of participants to treatment groups? |
| I 2. Was allocation to treatment groups concealed? |
| I 3. Were treatment groups similar at the baseline? |
| I 4. Were participants blind to treatment assignment? |
| I 5. Were those delivering treatment blind to treatment assignment? |
| I 6. Were outcomes assessors blind to treatment assignment? |
| I 7. Were treatment groups treated identically other than the intervention of interest? |
| I 8. Was follow up complete and if not, were differences between groups in terms of their follow up adequately described and analyzed? |
| I 9. Were participants analyzed in the groups to which they were randomized? |
| I 10. Were outcomes measured in the same way for treatment groups? |
| I 11. Were outcomes measured in a reliable way? |
| I 12. Was appropriate statistical analysis used? |
| I 13. Was the trial design appropriate, and any deviations from the standard RCT design (individual randomization, parallel groups) accounted for in the conduct and analysis of the trial? |

The Joanna Briggs Institute(JBI) critical appraisal checklist of the included randomized controlled trial studies

| Items(I) / Study | I1 | I2 | I3 | I4 | I5 | I6 | I7 | I8 | I9 | I10 | I11 | I12 | I13 | Total yes(%) | Risk of biais |
|---|---|---|---|---|---|---|---|---|---|---|---|---|---|---|---|
| First Author, year[Ref] | | | | | | | | | | | | | | | |
| | | | | | | | | | | | | | | | |
| | | | | | | | | | | | | | | | |
| | | | | | | | | | | | | | | | |

| Yes | + |
|---|---|
| No | - |
| Not Applicable | NA |
| Unclear | ? |

**Fig 1. Anticipated quality assessment of the included studies using the Joanna Briggs Institute critical appraisal checklist.**

## Timeline

We anticipate completing the search, screening, data extraction, and synthesis in late January 2025.

## Discussion

Recent research has focused on identifying barriers influencing adherence/compliance to Home-NIPPV in stable hypercapnic COPD patients [27,28]. The aim of this systematic review is to provide a comprehensive overview of the effectiveness of interventions targeting patient-related, device-related, and support service-related factors ((*e.g.,* place of initiation, ventilation mode type and settings, adjuvant therapies, and different types of monitoring) to enhance stable hypercapnic COPD patients' adherence/compliance to Home-NIPPV. We will conduct a systematic review of rigorous experimental studies to identify the most effective interventions and their efficacy in improving stable hypercapnic COPD patients' adherence/compliance. The findings of this systematic review are expected to contribute to a deeper scientific understanding of the topic, inform clinical practice and guide the development and implementation of more proactive, preventive, personalized, and participatory interventions to improve stable hypercapnic COPD patients' adherence/compliance to Home-NIPPV and, consequently outcomes in this population.

## Declaration

During the preparation of this work the authors used ChatGPT 3.5 in order to enhance the clarity and coherence of the manuscript' writing. The tool was utilized for language refinement purposes only, ensuring the text was clear and coherent without altering the scientific content or generating any new text. After using this tool, the authors reviewed and edited the content as needed and take full responsibility for the content of the publication [29,30].

## Supporting information

**S1 Checklist. PRISMA-P (Preferred Reporting Items for Systematic review and Meta-Analysis Protocols) 2015 checklist: Recommended items to address in a systematic review Protocol.**
(PDF)

## Acknowledgments

The authors would like to express their sincere gratitude to the reviewer for his/her excellent feedback, which has substantially improved the quality of this work. His/Her insightful comments and constructive suggestions were invaluable in refining our manuscript.

## Author contributions

**Conceptualization:** Amani Kacem, Imen Ben Saida, Helmi Ben Saad, Mohamed Boussarsar.

**Methodology:** Amani Kacem, Imen Ben Saida, Helmi Ben Saad.

**Writing – original draft:** Amani Kacem, Imen Ben Saida.

**Writing – review & editing:** Amani Kacem, Imen Ben Saida, Dhekra Chebil, Helmi Ben Saad, Mohamed Boussarsar.

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
