## [Decision Letter · Decision Letter 0]

15 Dec 2024

PONE-D-24-39536Interventions to improve adherence/compliance to home noninvasive positive pressure ventilation in stable hypercapnic Chronic Obstructive Pulmonary Disease patients : A Systematic Review ProtocolPLOS ONE

Dear Dr. BEN SAIDA,

Thank you for submitting your manuscript to PLOS ONE. After careful consideration, we feel that it has merit but does not fully meet PLOS ONE’s publication criteria as it currently stands. Therefore, we invite you to submit a revised version of the manuscript that addresses the points raised during the review process.

We look forward to receiving your revised manuscript.

Kind regards,

Felix Bongomin, MB ChB, MSc, MMed, FECMM

Academic Editor

PLOS ONE

Journal Requirements:

Reviewers' comments:

Reviewer's Responses to Questions

**Comments to the Author**

1. Does the manuscript provide a valid rationale for the proposed study, with clearly identified and justified research questions?

Reviewer #1: Yes

2. Is the protocol technically sound and planned in a manner that will lead to a meaningful outcome and allow testing the stated hypotheses?

Reviewer #1: Partly

3. Is the methodology feasible and described in sufficient detail to allow the work to be replicable?

Reviewer #1: Yes

4. Have the authors described where all data underlying the findings will be made available when the study is complete?

Reviewer #1: Yes

5. Is the manuscript presented in an intelligible fashion and written in standard English?

Reviewer #1: Yes

6. Review Comments to the Author

You may also provide optional suggestions and comments to authors that they might find helpful in planning their study.

Reviewer #1: Reviewer comments re: PONE-D-24-39536

In this study, the authors performed a systematic review protocol and on the Interventions to improve adherence/compliance to home noninvasive positive pressure ventilation in stable hypercapnic Chronic Obstructive Pulmonary Disease patients. This is a very important and relevant topic. The study protocol is well-designed and performed.

I feel however, there are some issues that need to be addressed before publication. I have included some suggestions for how the paper could be strengthened.

1. An assessment of the risk of bias in randomized controlled trials in the original studies was omitted. Review authors should categorize risk of bias as low, medium, or high. It would have been important for the review authors to assess the risk of bias among the included studies.

2. Searching exclusively in two databases may prove insufficient. I recommend that the author broaden the literature search by consulting additional databases beyond those currently utilized, as this may provide a more comprehensive understanding of the topic and facilitate the identification of a representative set of studies.

3. The methods section of the article currently lacks a subheading for "Data Synthesis." I kindly request that the author include this section, as data synthesis, whether through meta-analysis or qualitative synthesis, is a critical stage in the systematic review process. This stage involves pooling and evaluating the extracted data, which is essential for determining the outcomes of the review.

7. PLOS authors have the option to publish the peer review history of their article (what does this mean? ). If published, this will include your full peer review and any attached files.

**Do you want your identity to be public for this peer review?** For information about this choice, including consent withdrawal, please see our Privacy Policy .

Reviewer #1: No

---

## [Author Response · Author response to Decision Letter 0]

4 Jan 2025

Dear Editor-in-Chief of “Plos One”,

I would like to extend my gratitude to the editorial board and reviewers for their time and careful consideration of our manuscript entitled "Interventions to improve adherence/compliance to home noninvasive positive pressure ventilation in stable hypercapnic Chronic Obstructive Pulmonary Disease patients : A Systematic Review Protocol".

All necessary changes were brought to this manuscript, taking into account the comments of the reviewer.

Please note that we have acknowledged the reviewer in the acknowledgements section (please see lines 196-199)

We hope that this manuscript will be eligible for publication in your reputed journal.

Sincerely yours,

Reviewer comments for: Manuscript ID PONE-D-24-31948.

REFEREE 1

Comment :

In this study, the authors performed a systematic review protocol and on the Interventions to improve adherence/compliance to home noninvasive positive pressure ventilation in stable hypercapnic Chronic Obstructive Pulmonary Disease patients.

This is a very important and relevant topic.

The study protocol is well-designed and performed.

I feel however, there are some issues that need to be addressed before publication.

I have included some suggestions for how the paper could be strengthened.

Response :

Thank you for your thoughtful and encouraging feedback.

We greatly appreciate your recognition of the importance and relevance of this topic, as well as your positive remarks regarding the design and execution of our systematic review protocol.

Your suggestions for strengthening the paper are highly valued, and we have addressed the issues you raised to ensure the quality and impact of our work.

Thank you once again for your support and guidance!

Comment :

An assessment of the risk of bias in randomized controlled trials in the original studies was omitted. Review authors should categorize risk of bias as low, medium, or high.

It would have been important for the review authors to assess the risk of bias among the included studies.

Response :

Thank you for your valuable feedback.

We appreciate your observation regarding the “omission” of a risk of bias assessment in the RCTs included in our review protocol.

We agree that categorizing the risk of bias as low, medium, or high is essential for ensuring a comprehensive evaluation of the quality and reliability of the evidence.

 However, we would like to draw your attention to lines 145-154, where we have already included a paragraph stating: "Risk of bias (quality) assessment." Furthermore, we have included a figure (Figure 1) which illustrates the anticipated quality assessment of the included studies using the Joanna Briggs Institute critical appraisal checklist.

We acknowledge that there are many tools for assessing the risk of bias in RCTs in the original studies: e.g., Cochrane Risk-of-Bias Tool 2 (RoB 2), GRADE, JBI.

By systematically identifying and assessing potential biases, JBI critical appraisal tools for RCTs provide a valuable resource for researchers and clinicians to assess the quality of evidence and make informed decisions about clinical practice.

We acknowledge the robustness of other available tools, such as the widely used Cochrane Risk of Bias Assessment Tool. However, we have chosen the JBI Critical Appraisal Tools due to their simplicity and validated methodology. Furthermore, we are familiar with these tools and have prior experience in their application.

Thank you once again for your constructive input!

The following sentence exist in side the paper lines 145-154:

“Risk of bias (quality) assessment:

To assess the methodological quality of included studies, the Joanna Briggs Institute (JBI) critical appraisal tool will be used [25]. This tool evaluates various aspects of study design, with items rated as "yes," "no," "unclear," or "not applicable." To enhance reliability, two investigators (AK and IBS in the authors’ list) will independently apply the JBI checklist to each study. Discrepancies will be resolved through consensus, with a third investigator (DC in the authors’ list) mediating if necessary. A percentage score will be calculated based on the number of affirmative responses to checklist items. This checklist includes 13 items (Figure 1). Studies will be categorized as having high, moderate, or low risk of bias according to predetermined thresholds: ≤ 49%, 50-69%, and ≥ 70% affirmative responses, respectively [26].”

Figure 1 (see below) illustrates the anticipated quality assessment of the included studies using the Joanna Briggs Institute critical appraisal checklist.

Comment :

Searching exclusively in two databases may prove insufficient.

I recommend that the author broaden the literature search by consulting additional databases beyond those currently utilized, as this may provide a more comprehensive understanding of the topic and facilitate the identification of a representative set of studies

Response :

Thank you for your valuable feedback.

Searching at least 2 major databases is the minimum prerequisite for conducting a systematic review (Patole S, editor. Principles and Practice of Systematic Reviews and Meta-Analysis. Cham: Springer International Publishing; 2021. doi:10.1007/978-3-030-71921-0).

However, as recommended by the reviewer, expanding the search to include additional relevant databases can significantly enhance the comprehensiveness of the review.

We have added Cochrane Library to the 2 databases initially included.

As recommended by the reviewer, the following paragraph was added

lines 99-100:

“A comprehensive literature search will be conducted online, utilizing three reputable databases, PubMed, Scopus and Cochrane Library”

Lines 104-105:

“In the case of Scopus and Cochrane Library, the same terms will be sought in the article titles, abstracts, and/or keywords.”

Comment :

The methods section of the article currently lacks a subheading for "Data Synthesis."

I kindly request that the author include this section, as data synthesis, whether through meta-analysis or qualitative synthesis, is a critical stage in the systematic review process.

This stage involves pooling and evaluating the extracted data, which is essential for determining the outcomes of the review.

Response :

Thank you for your valuable feedback and for highlighting the importance of including a subheading for "Data Synthesis" in the Methods section.

We appreciate your suggestion, as data synthesis is indeed a crucial component of the systematic review process.

Thank you again for your constructive input, which will help improve the clarity and comprehensiveness of the manuscript.

We have included a dedicated subsection titled "Data Synthesis" within the Methods section to outline the approach used for pooling and evaluating the extracted data.

Please see lines 157-173.

---

## [Editor Report · Decision Letter 1]

15 Jan 2025

Interventions to improve adherence/compliance to home noninvasive positive pressure ventilation in stable hypercapnic chronic obstructive pulmonary disease patients : A systematic review protocol

PONE-D-24-39536R1

Dear Dr. BEN SAIDA,

We’re pleased to inform you that your manuscript has been judged scientifically suitable for publication and will be formally accepted for publication once it meets all outstanding technical requirements.

Kind regards,

Felix Bongomin, MB ChB, MSc, MMed, FECMM

Academic Editor

PLOS ONE
---

## [Editor Report · Acceptance letter]

PONE-D-24-39536R1

PLOS ONE

Dear Dr. Ben Saida,

I'm pleased to inform you that your manuscript has been deemed suitable for publication in PLOS ONE. Congratulations! Your manuscript is now being handed over to our production team.

Kind regards,

on behalf of

Dr. Felix Bongomin

Academic Editor

PLOS ONE